# Prognostic Impact of EBUS TBNA for Lung Adenocarcinoma Patients with Postoperative Recurrences

**DOI:** 10.3390/diagnostics12102547

**Published:** 2022-10-20

**Authors:** Ying-Yi Chen, Ying-Shian Chen, Tsai-Wang Huang

**Affiliations:** 1Graduate Institute of Medical Science, National Defense Medical Center, Taipei 11490, Taiwan; 2Division of Thoracic Surgery, Tri-Service General Hospital, National Defense Medical Center, Taipei 11490, Taiwan; 3Division of Thoracic Surgery, Taichung Armed Forced General Hospital, Taichung 411228, Taiwan

**Keywords:** endobronchial ultrasound in transbronchial biopsy, lung adenocarcinoma

## Abstract

Background: The aim of this study was to verify the importance and the timing of endobronchial ultrasound with transbronchial biopsy (EBUS TBNA) among lung adenocarcinoma patients after radical resection. Methods: We retrospectively reviewed consecutive patients with non-small cell lung cancer (NSCLC) who had ever received radical resection from January 2002 to December 2021. The patients were divided into two groups, with and without EBUS TBNA, for diagnosis or staging. Results: Of 2018 patients with NSCLC, after surgical resection of lung tumors, there were 232 with recurrences. Under multivariate Cox regression analysis, patients with recurrences who received EBUS TBNA had a statistically higher mean maximum standardized uptake value (SUVmax) (hazard ratio (HR) = 1.115, confidence interval (CI) = 1.004–1.238, *p* = 0.042) and better survival (HR = 5.966, CI = 1.473–24.167, *p* = 0.012). Although KM survival analysis showed no statistically significant difference between groups with and without EBUS TBNA (*p* = 0.072) of lung adenocarcinoma patients with recurrences, patients with mutated epidermal growth factor receptor (EGFR) showed significantly better survival than wild-type EGFR (*p* = 0.007). Conclusions: The clinical practice of EBUS TBNA is not only for diagnosis, but also for nodal staging. We found that lung adenocarcinoma patients with recurrences who received EBUS TBNA had better overall survival. Therefore, EBUS TBNA is a reliable and feasible tool that could be used in lung adenocarcinoma patients with recurrences for early diagnosis and for adequate tissue specimens for further molecular analysis.

## 1. Introduction

In the large series on postresection recurrence of non-small cell lung cancer (NSCLC), nodal status was independently associated with local and distant recurrence [1], and the number of metastatic lymph nodes (LNs) had a strong impact on survival, in addition to the current nodal staging classification [2]. Therefore, precise diagnosis of nodal status is quite important for appropriate treatment, especially in patients with recurrences. Endobronchial ultrasound-guided transbronchial needle aspiration (EBUS TBNA) is recommended as a minimally invasive procedure compared with mediastinoscopy for lung cancer staging and diagnosis [3,4,5,6]. In the ESMO clinical practice guidelines for diagnosis, treatment and follow-up, EBUS-TBNA is recommended as the most common diagnostic test for central tumors or regional lymph nodes [7]. In a recent prospective multicentric study [8], EBUS-TBNA showed a higher diagnostic yield when compared to any other bronchoscopic sampling technique and was independently associated with a higher probability of diagnosis at multivariate analysis. The combination (CUS) of EBUS-TBNA and endoscopic ultrasound-fine needle aspiration (EUS-FNA) allows complete staging of the mediastinum in NSCLC patients [9]. Based on the improvements in personalized medicine, EBUS-TBNA is also a less invasive way of providing adequate samples for molecular tests [10]. Long-term overall survival significantly differed according to nodal stages in NSCLC, highlighting the importance of EBUS TBNA [11]. To the best of our knowledge, long-term survival analyses of clinical nodal stage diagnosis by EBUS TBNA in recurrent lung adenocarcinoma have not been reported. Although the algorithm for locoregional lymph-node staging in patients with non-metastatic NSCLC is established [7], the role of EBUS TBNA in recurrent lung adenocarcinoma still needs further investigation. Therefore, this study aimed to verify the prognostic impact of EBUS TBNA in lung adenocarcinoma patients with postoperative recurrences.

## 2. Materials and Methods

### 2.1. Patient Selection and Study Design

The database of the Thoracic Surgery Division of Tri-Service General Hospital, National Defense Medical Center, Taipei, Taiwan, was searched for patients who underwent surgical resection (wedge resection, segmentectomy, lobectomy, pneumonectomy) for NSCLC between January 2002 and December 2021. The medical records of the study population were reviewed and evaluated. All patients with pathologic stage III and IV received adjuvant chemotherapy. Patients who received EBUS TBNA for their diagnosis were suspected to have intrathoracic lymph node metastases based on enlargement (short axis > 10 mm) visualized by computed tomography (CT) or F-18 fluorodeoxyglucose (FDG) uptake ≥ standard uptake value (SUV) 3.5 on positron emission tomography (PET) scans. All LNs in the thorax and extrathoracic regions with SUVmax >3.5 were considered positive, unless they showed high attenuation (>70 HU) or benign calcification (central nodular, laminated, popcorn or diffuse) on the soft-tissue window of their respective CT images [12,13,14]. All patients with recurrences had completed a PET-CT scan. This study was approved by the Institutional Review Board of Tri-Service General Hospital, National Defense Medical Center (2-108-05-089). Written informed consent for bronchoscopy was obtained from all patients; additional informed consent for this study was waived due to the design of retrospective chart review for clinical history and diagnostic results. Chest CT every 6 months until 2 years postoperatively and once a year thereafter was used as the standard follow-up protocol. PET scan and brain surveillance were not mandatory; introductions of these depended on physicians. The initial sites of recurrence were defined as the recurrent sites diagnosed first after resection; other sites were identified in subsequent systemic surveillance and those diagnosed within 1 month of the initial detection. All patients with recurrences received a PET-CT scan. The SUVmax values of mediastinal lymph nodes were recorded for our comparative analysis.

### 2.2. Mediastinal Lymph Node Sampling

EBUS TBNA was performed in an operation room by two pulmonologists under general anesthesia with 8.0mm endotracheal intubation. The procedure has been carried out in this manner since January 2015. After white-light bronchoscopy was performed through a tracheal tube, the target lymph nodes and peripheral vessels were examined by EBUS using a linear-array ultrasonic bronchoscope (BF-UC180F-OL8; Olympus Ltd., Tokyo, Japan). The diameter of the target lymph nodes was measured and recorded on a frozen ultrasound image. A dedicated 22 G needle was used for aspiration (NA-201SX-4022; Olympus Ltd. Tokyo, Japan). We recommend that at least two needle aspirations be performed for each target lesion, and the number of moves of each pass was about twenty [15,16]. All procedures were conducted by experienced bronchoscopists. An internal stylet was removed after the initial puncture and negative pressure, was applied with a syringe to obtain histological cores and cytological specimens. The aspirated material was smeared onto glass slides; smears were fixed in 95% alcohol. Papanicolaou staining and light microscopy were also performed by an independent cytopathologist. Histological cores were fixed with formalin and stained with hematoxylin and eosin. Immunohistochemistry was also performed when necessary. Biopsies from stations 2R, 2L, 10R, 10L, 4R, 4L and 7 were routinely obtained. The EGFR mutation status for each patient was obtained using tumor specimens from diagnostic or surgical procedures by EBUS TBNA or surgical resection. Sequencing of epidermal growth factor receptor exons 18 to 21 was performed per the institutional pathology laboratory protocol using the Sanger technique, as previously described [17]. In lung adenocarcinoma patients with recurrences, EBUS TBNA was performed after radical resection for diagnosis and re-staging.

### 2.3. Statistical Analyses

Kaplan–Meier curves were utilized to estimate survival distributions. Time-to-event comparisons were performed using log-rank tests. Univariate and multivariate analyses assessed the clinicopathologic factors of NSCLC patients with and without EBUS TBNA. A *t*-test was used for the comparison of continuous variables, and the Chi-square test or Fisher’s exact test, when appropriate, was used for categorical variables. All reported *p* values are two-sided, and no adjustments have been made for multiple comparisons. Significant variables in univariate analysis or those deemed clinically important were then entered into a multivariable logistic regression model. The IBM SPSS Statistics for Windows software package (version 22.0; IBM Corp., Armonk, NY, USA) was used for the data analysis.

The following baseline patient or tumor parameters were analyzed for this study: age at diagnosis of lung cancer, gender, race and self-reported smoking status—prospectively collected; pathology staging; presence of extrathoracic tumor at the diagnosis of metastatic disease; metastatic site(s) (characterized at up to 1 month within start of systemic therapy for the liver, adrenals, bone, brain and leptomeninges); and type of sensitizing EGFR mutation. Smoking status was classified as never (<100 lifetime cigarettes), former (quit ≥1 year before start of therapy) or current (active or quit within 1 year prior to start of therapy). OS was calculated from the date of start of radical resection or the first-line systemic treatment for lung cancer until death from any cause. Patients still alive were censored at their last follow-up visit. Disease progression was defined as the date of radiographic imaging which demonstrated progression deemed clinically significant by the physician—whether due to resultant patient symptoms or due to a radiographic change that was significant enough to warrant a discussion of change in therapy [18]. Recurrences of lung cancer were diagnosed after surgical biopsies. If there was no evidence of N2 or N3 disease on the chest CT and PET-CT scan, patients underwent surgical resection of the tumor. Pulmonary resection and a systematic nodal dissection were performed in every patient by a thoracotomy or video-assisted thoracic surgery. If there was suspicion of nodal metastases, EBUS TBNA would be arranged, with surgery at the same time or at a different time. The results of the surgical pathology were seen as definitely diagnostic results.

## 3. Results

### 3.1. Demographic Characteristics of the Enrolled Patients

A total of 2018 patients with NSCLC were eligible for this retrospective cohort study, and seventy-nine of them received EBUS TBNA for staging or diagnosis over the period of January 2002–February 2021. We selected 232 lung adenocarcinoma patients with recurrences for further investigation (Table 1), which showed patients who received EBUS TBNA had significantly better survival (*p* = 0.002), higher SUVmax for recurrent tumors (*p* = 0.018) and more dissected lymph nodes (*p* = 0.022). There were no statistical significances in age (*p* = 0.372), gender (*p* = 0.359), operation type (*p* = 0.586), differentiation (*p* = 0.19), EGFR mutation (*p* = 0.196), location (*p* = 0.633), smoking habits (*p* = 0.522), lymphovascular space invasion (LVSI) (*p* = 0.289), VPI (*p* = 0.348), pathologic stages (*p* = 0.72), tumor size (*p* = 0.373), carcinoembryonic antigen (CEA) (*p* = 0.67) and GGO ratio (*p* = 0.443). No endobronchial mucosal abnormality was found in any patient. All EBUS TBNA procedures were performed under general anesthesia by two experienced pulmonologists. Each node underwent a median of three passes (range: 2–5). The mean number of mediastinal lymph node stations biopsied per patient was 1.7 (range: 1.0–2.3).

### 3.2. Univariate and Multivariate Analysis of Predictive Factors for Prognostic Impact of EBUS TBNA in Lung Adenocarcinoma Patients with Recurrences

Table 2 shows the results of univariate regression and multivariate regression analyses. Under univariate analysis, patients who had ever received EBUS TBNA for recurrent lung adenocarcinoma had significantly more dissected lymph nodes (HR = 1.068, *p* = 0.028), SUVmax for recurrent tumors (HR = 1.113, *p* = 0.025) and survival (HR = 6.202, *p* = 0.012). Furthermore, the SUVmax for recurrent tumors (HR = 1.115, *p* = 0.042) and survival (HR = 5.966, *p* = 0.012) by multivariate Cox regression analysis showed statistically significant differences.

### 3.3. Survival Analysis

The Kaplan–Meier analysis revealed no significant differences (*p* = 0.072) in overall survival (OS) between the two groups, with and without EBUS TBNA (Figure 1). The group with EBUS TBNA did not all survive, but the mean OS was better than that of the other group. We further investigated the influence of mutated EGFR. In the group without EBUS TBNA, patients with mutated EGFR showed significantly better survival (*p* = 0.007). Patients who received EBUS TBNA were not statistically different in OS from patients with wild-type and mutated EGFR (*p* = 0.587).

## 4. Discussion

A precise assessment of lymph-node metastasis is important for deciding on the optimal treatment for patients with NSCLC. To the best of our knowledge, predictive factors of lymph node metastasis, preoperatively, are central tumor localization [19], larger tumor size [19,20,21], age ⩽ 67 years [20], high CEA level [20,21], micropapillary predominant adenocarcinoma [21,22,23] and consolidation/tumor ratio ≥ 89% [20]. EBUS TBNA is recommended as a more feasible and convenient procedure than mediastinoscopy and video-assisted thoracoscopic surgery with lymph node dissection. Rapid onsite evaluation (ROSE) is a tool that is believed to increase the adequacy rate, diagnostic yield and accuracy of EBUS TBNA [24,25,26,27]. According to a literature review, Kim J. et al. [28] showed high diagnostic value and high suitability for EGFR mutation analysis with regard to re-biopsy in patients with previously treated lung cancer. Sanz-Santos J. et al. [29] proved EBUS-TBNA is an accurate procedure for the diagnosis of locoregional recurrence of surgically treated lung cancer. Although they all investigated the role of EBUS TBNA in recurrent lung cancer, no associated outcomes were discussed. Although the algorithm of EBUS TBNA for non-metastasized NSCLC was established in patients with de novo treatment [7], the role of EBUS TBNA in recurrent lung adenocarcinoma still requires further investigation.

### Predictive Factors of the Prognostic Impact of EBUS TBNA for Recurrent Lung Adenocarcinoma

EGFR mutations are strongly associated with clinical outcomes in patients with lung adenocarcinoma. However, the prognosis of EGFR mutation status is still equivocal [30,31,32,33,34,35]. In our study, age (*p* = 0.372), gender (*p* = 0.359), smoking habits (*p* = 0.522), operation (*p* = 0.586), tumor differentiation (*p* = 0.19), EGFR mutation status (*p* = 0.196), location (*p* = 0.633), CEA level (*p* = 0.67), LVSI (*p* = 0.289), VPI (*p* = 0.348), pathologic stage (*p* = 0.72), tumor size (*p* = 0.373) and GGO ratio (*p* = 0.443) were not statistically significant between the groups with and without EBUS TBNA for recurrent lung adenocarcinoma. We found a statistically significantly higher SUVmax for recurrent tumors and more dissected lymph nodes by radical resection in patients who underwent EBUS TBNA for recurrent lung adenocarcinoma. The OS of recurrent lung adenocarcinoma patients was better when they underwent EBUS TBNA. The above result is compatible with clinical practice and reveals that EBUS TBNA could improve the accuracy of nodal status for patients with recurrent lung adenocarcinoma.

In the univariate regression analysis, the SUVmax of recurrent tumors, number of dissected lymph nodes and survival were significant predictors for the prognostic impact of EBUS TBNA. After multivariate analysis for recurrent lung adenocarcinoma, SUVmax of recurrent tumors was an independent predictor for patients who received EBUS TBNA of higher survival. High FDG uptake in a PET scan that shows possible nodal metastases needs histopathologic confirmation and represents higher metabolic activity [36]. However, inflammation or infection of mediastinal lymph nodes would give false positive results in a PET–CT scan. Precise diagnosis of nodal status is more certain by EBUS TBNA. Although the behavior of nodal status in recurrent lung adenocarcinoma could be correlated with the expression of the FDG uptake value, tissue sampling of mediastinal lymph nodes by EBUS TBNA for nodal status is still recommended as a more accurate modality. Therefore, EBUS TBNA should be wildly applied to patients with higher SUVmax values of recurrent tumors for improving OS.

For non-metastasized and untreated NSCLC, EBUS TBNA is recommended as a useful modality for nodal staging [7]. Doctor Piergiorgio Muriana [37] published a review on the role of EBUS TBNA in lung cancer restaging and mutation analysis and suggested EBUS TBNA should be used in patients with NSCLC who need a restaging of disease after induction therapy, or show progression in the course of therapy with tyrosine kinase inhibitors (TKIs) or immune therapy, to guide subsequent treatments. However, there are not many studies on nodal status for recurrent lung adenocarcinoma. In our hypothesis, nodal status is important for lung adenocarcinoma patients with recurrences. According to Figure 1, OS in the group with EBUS TBNA was better than the group without EBUS TBNA, although there was no statistical significance. Furthermore, Figure 2 showed patients with mutated EGFR had significantly better survival than patients with wild-type EGFR in the group without EBUS TBNA. Moreover, the impact of EBUS TBNA on patients with wild-type EGFR was not significantly different from that of EBUS TBNA on patients with mutated EGFR. EBUS TBNA for recurrent lung adenocarcinoma could eliminate the difference in OS, especially for patients with the wild type.

A substantial (40–60%) proportion of patients with NSCLC have EGFR mutations. Treatment strategies for patients with advanced-stage NSCLC have markedly changed, from the empirical use of cytotoxic agents to targeted regimens, such as EGFR TKIs. EGFR TKIs, the first-line therapy for advanced NSCLC, are reported to be the most effective [38]. Recently, the FLAURA study [39], using the third-generation EGFR-TKI osimertinib, demonstrated an OS extension by a median of 6.8 months compared with standard EGFR-TKIs, along with a 20% reduction in the risk of mortality. Osimertinib was also shown to lead to a statistically significant reduction in the risk of central nervous system disease progression. In addition, 28% of patients remained on osimertinib treatment for 3 years, considerably longer than those in the comparator group (9%) [39]. The duration of the first subsequent treatment with osimertinib was 25.5 months, compared with 13.7 months with standard EGFR-TKIs [39]. Thus, the long-term OS benefit with first-line osimertinib highlights a promising option in the management of stage IV NSCLC [39]. According to our data, different regimens of TKIs might influence OS in both groups. No matter what regimen of TKIs patients received, their survival was still better than patients with wild-type EGFR. Despite this, the use of EBUS TBNA could help to increase survival in recurrent lung adenocarcinoma patients with wild-type EGFR.

However, the replacement of mediastinoscopy with EBUS TBNA in lung cancer staging should be based on the quality and diagnostic sensitivity of EBUS TBNA [40]. Our previous prospective studies obtained fairly acceptable sensitivities for EBUS TBNA staging or diagnosis in patients with all stages of NSCLC (positive predictive rate was 97.67% and accuracy rate was 77.38%) [41]. The role of technical features involved in EBUS TBNA outcomes in the search for molecular aberrations has been widely investigated. Several studies pointed out that the choice of needle, number of passes, use of ROSE, sample cellularity, sample contamination by surrounding necrosis or blood elements and sample processing are determinant factors for obtaining suitable material [42]. The CHEST guidelines for EBUS TBNA released in 2016 recommend—regardless of ROSE availability—at least three passes for each sampled station, and possibly additional passes to increase the effectiveness of mutation analysis, but with a low level of evidence [43]. In the present study, we targeted an average of 1.7 mediastinal stations per patient and performed at least two aspirations per target.

The limitations of this study were its retrospective and single-center design. Moreover, the small sample size may have caused selection bias. We used IBM SPSS for evaluation of test power, and the power was 70.7% (<80%). A larger number of cases in the EBUS TBNA group is needed. Although all patients with pathological stage III and IV received adjuvant chemotherapy, these patients received different regimens of TKIs, chemotherapy and immune therapy for recurrences. Therefore, the impacts of different therapies for recurrences were not investigated. EBUS TBNA was performed under general anesthesia through an endotracheal tube in all cases. This might have contributed to the high diagnostic yield in this study compared with awake patients. However, stations 2R and 2L were sometimes difficult to assess because of the presence of the endotracheal tube. Furthermore, different cytopathologists were present for ROSE and EBUS TBNA. ROSE has been shown to reduce the number of TBNAs necessary for a firm diagnosis [44]. ROSE is believed to be a useful tool for EBUS TBNA to increase diagnostic accuracy. However, false negative results could not be totally ruled out. Therefore, further invasive procedures, with mediastinoscopy or VATS with mediastinal lymph node dissection, were still necessary in some cases. In our study, EBUS TBNA was performed by experienced thoracic surgeons with extensive familiarity with mediastinal anatomy, and correlated with radiologic findings. Thus, the excellent results obtained may not be generalizable to all studies of EBUS TBNA. Finally, preoperative staging before EBUS TBNA and mediastinoscopy was mainly based on CT findings, because PET scanning was not available for all patients at the start of the study. By combining CT and PET for noninvasive mediastinal lymph-node staging, clinical staging could have been used in our study, as both techniques have equivalent accuracy in the mediastinal staging of lung cancer. The accuracies of chest CT and PET-CT scans should also be investigated and compared with that of EBUS TBNA in further studies.

## 5. Conclusions

The prognostic use of EBUS TBNA in recurrent lung adenocarcinoma was closely related to high OS, especially in patients with higher SUVmax for recurrent tumors and more dissected lymph nodes. Long-term survival differed according to nodal status by EBUS TBNA in recurrent lung adenocarcinoma, highlighting the importance of the use of EBUS TBNA, particularly for the differentiation of wild-type and mutated EGFR. The reason for the high OS in patients with recurrent lung cancers might be early diagnosis and the acquisition of adequate tissue specimens for further molecular analysis. A larger number of cases in the EBUS TBNA group is needed.

## Figures and Tables

**Figure 1 diagnostics-12-02547-f001:**
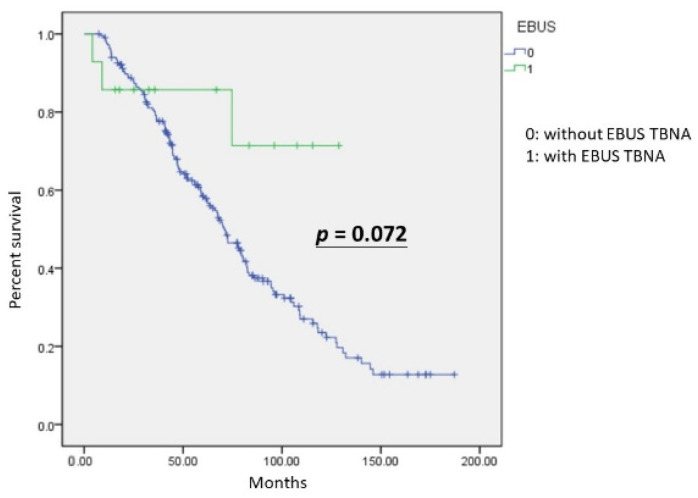
The Kaplan–Meier curve was not statistically significant for overall survival in recurrent lung adenocarcinoma with and without EBUS TBNA (*p* = 0.072).

**Figure 2 diagnostics-12-02547-f002:**
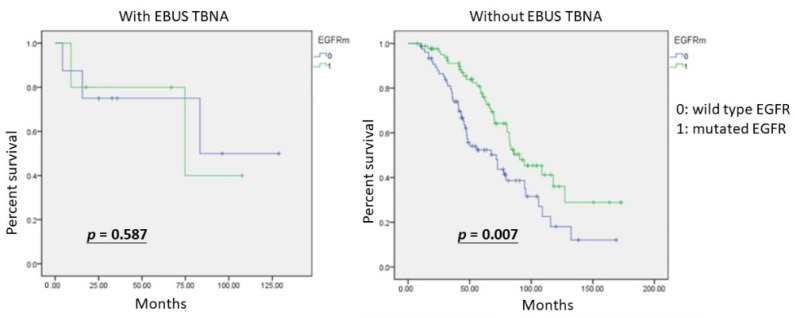
The Kaplan–Meier curves for overall survival of recurrent lung adenocarcinoma patients treated with EBUS TBNA are not statistically significantly different for the mutated EGFR and wild-type EGFR (*p* = 0.587). In contrast, overall survival in recurrent lung adenocarcinoma patients without EBUS TBNA treatment different statistically significantly between those with mutated EGFR and those with wild-type EGFR (*p* = 0.007).

**Table 1 diagnostics-12-02547-t001:** The postoperative comparison of lung adenocarcinoma patients with recurrences by EBUS TBNA for diagnosis and staging, or not.

	Recurrent Lung Adenocarcinoma without EBUS TBNA n = 218 (%)	Recurrent Lung Adenocarcinoma with EBUS TBNA n = 14 (%)	*p*-Value ^a^
Gender			0.359
Male	97 (44.49)	8 (57.14)	
Female	121 (55.51)	6 (42.86)	
Operation			0.586
Wedge	32 (14.68)	1 (7.14)	
Segmentectomy	6 (2.75)	1 (7.14)	
Lobectomy	180 (82.57)	12 (85.71)	
Differentiation			0.190
Well	47 (21.56)	2 (14.29)	
Moderate	116 (53.21)	6 (42.86)	
Poor	55 (25.23)	6 (42.86)	
EGFR			0.196
Exon 18 mutation	1 (0.63)	0	
Exon 19 deletion	47 (29.38)	2 (14.29)	
L858R	34 (21.25)	3 (21.43)	
Exon 20 mutation	8 (5)	0	
Wild-type	70 (43.75)	9 (64.29)	
Location			0.633
Central	76 (34.86)	4 (28.57)	
Peripheral	142 (65.14)	10 (71.43)	
Smoking			0.522
Yes	75 (34.4)	6 (42.86)	
No	143 (65.6)	8 (57.14)	
Survival			**0.002** ^a^
Yes	81 (37.16)	11 (78.57)	
No	137 (62.84)	3 (21.43)	
LVSI			0.289
Absent	178 (81.65)	13 (92.86)	
Present	40 (18.35)	1 (7.14)	
VPI			0.348
Absent	202 (92.66)	12 (85.71)	
Present	16 (7.34)	2 (14.29)	
p-stage			0.720
I	115 (52.75)	9 (64.29)	
II	43 (19.72)	1 (7.14)	
III	46 (21.11)	3 (21.43)	
IV	14 (6.42)	1 (7.14)	
Age (year)	61.43 ± 10.92	64.07 ± 6.72	0.372
SUVmax of recurrent tumors	6.13 ± 4.52	9.56 ± 8.07	**0.****018** ^b^
Tumor size (cm)	2.7 ± 1.3	2.38 ± 1.26	0.373
CEA (ng/mL)	8.16 ± 17.61	10.24 ± 17.35	0.670
Dissected lymph nodes	12.06 ± 6.95	16.86 ± 13.83	**0.****022** ^b^
GGO ratio	0.19 ± 0.27	0.14 ± 0.15	0.443

^a^ Significance was assessed using χ^2^ test. ^b^ Significance was assessed using Student’s *t*-test. Key: SCC, squamous cell carcinoma; LVSI, lymphovascular space invasion; p-stage, pathologic stage; VPI, visceral pleural. invasion; SUVmax, maximum standard uptake value of FDG; CEA, carcinoembryonic antigen; GGO, ground-glass opacity.

**Table 2 diagnostics-12-02547-t002:** Univariate and multivariate logistic regression analysis of clinicopathologic factors for recurrent lung adenocarcinoma by EBUS TBNA.

	Univariant		*p*-Value ^a^	Multi-Variant		*p*-Value ^a^
	HR	CI (95%)		HR	CI (95%)	
Number of dissected lymph nodes	1.068	1.007–1.132	**0.028** ^a^	1.041	0.978–1.108	0.209
SUVmax of recurrent tumors	1.113	1.013–1.223	**0.025** ^a^	1.115	1.004–1.238	**0.042** ^a^
Survival	6.202	1.68–22.889	**0.012** ^a^	5.966	1.473–24.167	**0.012** ^a^

^a^ Significance was assessed using Student’s *t*-test. Key: HR, hazard ratio; CI, confidence interval; SUVmax, maximum standard uptake value of FDG.

## Data Availability

Not applicable.

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
