# Peer review of "Prognostic Impact of EBUS TBNA for Lung Adenocarcinoma Patients with Postoperative Recurrences"

_diagnostics, 2022, doi:10.3390/diagnostics12102547_

Round 1

Reviewer 1 Report (New Reviewer)

The paper is retrospective but very interesting. Lung cancer recurrence is a topic of recent interest, especially with the advent of new therapy targets. The usefulness of EBUS TBNA is not new by now, but in this field, as the authors say, there are broad prospects for evaluation.

Two similar works published in 2017 and 2019 which took into consideration not only surgical patients, not mentioned in the bibliography [Kim J.et al Cancer Res Treat. 2019; 51 (4): 1488-1499 AND Sanz-Santos J et al BMC Pull Med 2017 Feb 28;17(1):46]. I believe the two works need to be mentioned.

Author Response

Thanks for your suggestions. We added these two references in the discussion section (line 192-197).

Reviewer 2 Report (New Reviewer)

The manuscript reports the prognostic impact of EBUS TBNA in postoperative recurrence. The authors conclude that EBUS TBNA is a reliable and feasible tool to be used in lung adenocarcinoma patients with recurrences. However, the study is insufficient to state this conclusion.

The number of EBUS-TBNA groups is too small for statistical analysis.

If the authors wish to demonstrate this conclusion, a larger number of cases in the EBUS TBNA group is needed. In addition, the test power needs to be calculated.

Author Response

Thanks for your suggestions. We used IBM SPSS for evaluation test power and the power was 0.707 (70.7%). We would add the sentence “a larger number of cases in the EBUS TBNA group is needed” in the conclusion (line 273-275, 304-305).

Reviewer 3 Report (New Reviewer)

Article written in an unclear way both for the methodology and for the study objectives. The analyzed topic is interesting and of great scientific interest but many points should be improved:

1) line 56: is a wedge resection considered radical resection? modify this sentence in order not to make mistakes for the reader.

2) who is doing the ebus-tbna or tbna procedure? the surgeon or the pulmonologist? in line 78 it speaks of surgeon, in line 144 it speaks of two expert pulmonologists

3) The study groups do not seem homogeneous: 14 patients undergoing Ebus-Tbna vs 218 undergoing Tbna. The ebus study group of only 14 patients is too small to be able to analyze the data in a statistically significant way.

Furthermore,  it is importante to explain better what and how it is analyzed especially in relation to the SUV and EGFR. The reasoning is not logical and the reader has to put all the pieces together in disorder to arrive at a conclusion.

Author Response

Point 1: line 56: is a wedge resection considered radical resection? modify this sentence in order not to make mistakes for the reader.

Response 1: Thanks for your suggestions. We would revise it as surgical resection (line 56).

Point 2: Who is doing the ebus-tbna or tbna procedure? the surgeon or the pulmonologist? in line 78 it speaks of surgeon, in line 144 it speaks of two expert pulmonologists

Response 2: Thanks for your suggestion. Actually, all EBUS TBNA were perfomed by two experienced pulmonologists and we were also thoracic surgeons. We were glad to revise that as the same title (line 79).

Point 3: The study groups do not seem homogeneous: 14 patients undergoing Ebus-Tbna vs 218 undergoing Tbna. The ebus study group of only 14 patients is too small to be able to analyze the data in a statistically significant way.

Response 3: Thanks for your suggestions. We used IBM SPSS for evaluation test power and the power was 0.707 (70.7%). We would add the sentence “a larger number of cases in the EBUS TBNA group is needed” in the conclusion (line 273-275, 304-305).

Point 4: Furthermore, it is important to explain better what and how it is analyzed especially in relation to the SUV and EGFR. The reasoning is not logical and the reader has to put all the pieces together in disorder to arrive at a conclusion.

Response 4: Thanks for your suggestions. Although the sample size of our study is small, the aim of the study is to investigate the importance of EBUS TBNA in recurrent lung adenocarcinoma for evaluation of surgical outcomes. The reason of high OS in recurrent lung adenocarcinoma is the general advantages of EBUS TBNA for clinical practice. And we found the risk factors in recurrent lung adenocarcinoma by EBUS TBNA were high SUVmax and more mediastinal lymph node dissection.

Reviewer 4 Report (New Reviewer)

This manuscript reports on a retrospective study investigating the impact of EBUS-TBNA in patients with lung adenocarcinoma who experienced recurrence after surgery, comparing survival in 218 patients who did not receive EBUS-TBNA and 14 patients who did. However, there are several problems with comparing these populations. First, the small number of patients who underwent EBUS-TBNA (14 patients) makes it difficult to draw conclusions, especially with many censored cases. Second, the backgrounds of the patients are different in the two groups, making comparisons of survival not meaningful. To compare survival rates in both groups, the authors need to adjust for background. In particular, for the pathologic stage, the authors argued that there was no significant difference, but this is due to the small sample size. In addition, a multivariate analysis should be performed if the number of events was sufficient. Finally, the comparison is essentially irrelevant because the EBUS-TBNA itself at the time of diagnosis may alter the clinical stage and treatment. Rather, I would suggest combining patients who relapsed and those who did not and comparing survival rates to see the impact of EBUS-TBNA. 

Author Response

Point 1: First, the small number of patients who underwent EBUS-TBNA (14 patients) makes it difficult to draw conclusions, especially with many censored cases.

Response 1: Thanks for your suggestions. We used IBM SPSS for evaluation test power and the power was 0.707 (70.7%). We would add the sentence “a larger number of cases in the EBUS TBNA group is needed” in the conclusion (line 273-275, 304-305).

Point 2: Second, the backgrounds of the patients are different in the two groups, making comparisons of survival not meaningful. To compare survival rates in both groups, the authors need to adjust for background. In particular, for the pathologic stage, the authors argued that there was no significant difference, but this is due to the small sample size. In addition, a multivariate analysis should be performed if the number of events was sufficient.

Response 2: Thanks for your suggestions. The pathologic stage of two groups differed largely, but there was still no statistical significance. Small smaple size is the main cause. We also perfomed multivariate analysis to evaluate the significance of predictors in table 2. We revised the presentation of pathologic stage in table 1. Thanks for your help.

Point 3: Finally, the comparison is essentially irrelevant because the EBUS-TBNA itself at the time of diagnosis may alter the clinical stage and treatment. Rather, I would suggest combining patients who relapsed and those who did not and comparing survival rates to see the impact of EBUS-TBNA. 

Response 3: Thanks for your question. However, lung adenocarcinoma patients without recurrences did not receive EBUS TBNA for regular follow up in clinical practice of cancer surveillance. Therefore, we think it might be no beneficial role in lung adenocarcinoma patients without recurrences. EBUS TBNA is also an invasive procedure than imaging modality.

Reviewer 5 Report (New Reviewer)

In general, the manuscript "Prognostic impact of EBUS TBNA for lung adenocarcinoma patients with postoperative recurrences" is well written scientifically and regarding format.

Comments:

1. Did the patients included in the study (in EBUS-TBNA group and non EBUS-TBNA group) receive any pre- or postoperative radiotherapy, chemotherapy, immunotherapy and/or targeted treatment before the recurrence? It can impact the yield of restaging EBUS-TBNA or other means of biopsy procedures. This should be clearly given in Methods. There should also be a comment on this issue in Discussion.

2. The EBUS-TBNA group is quite smaller than the non EBUS-TBNA group. This can have impact on the results and conclusion of the study. The authors should comment on this issue in the Discussion.

3. English should be carefully improved in some parts of the text.

Author Response

Point 1: Did the patients included in the study (in EBUS-TBNA group and non EBUS-TBNA group) receive any pre- or postoperative radiotherapy, chemotherapy, immunotherapy and/or targeted treatment before the recurrence? It can impact the yield of restaging EBUS-TBNA or other means of biopsy procedures. This should be clearly given in Methods. There should also be a comment on this issue in Discussion.

Response 1: Thanks for your suggestions. All stage III and IV patients received postoperative adjuvant chemotherapy before recurrences. We added this explanation in Methods (line 58-59) and Discussion (line 275-277).

Point 2: The EBUS-TBNA group is quite smaller than the non EBUS-TBNA group. This can have impact on the results and conclusion of the study. The authors should comment on this issue in the Discussion.

Response 2: Thanks for your suggestions. We used IBM SPSS for evaluation test power and the power was 0.707 (70.7%). We would add the sentence “a larger number of cases in the EBUS TBNA group is needed” in the conclusion (line 273-275, 304-305).

Point 3: English should be carefully improved in some parts of the text.

Response 3: Thanks for your suggestions. We would keep English editing. Thanks for your help.

Round 2

Reviewer 2 Report (New Reviewer)

Thank you for the re-submission.

The manuscript has been much improved and is in a nice condition now.

Reviewer 3 Report (New Reviewer)

All corrections have been made. Overall, the article is still a bit difficult and not very smooth for the reader but in the end the message is clear and functional.

Reviewer 4 Report (New Reviewer)

No further comments.

This manuscript is a resubmission of an earlier submission. The following is a list of the peer review reports and author responses from that submission.

Round 1

Reviewer 1 Report

EBUS-TBNA is an integral tool in the diagnosis and staging of lung cancer and other diseases involving mediastinal lymphadenopathy. This work aimed to verify the importance and the timing of EBUS TBNA among lung adenocarcinoma patients after radical resection. In my oppinion this manuscript does not bring anything groundbreaking to the current state of knowledge/

Additionally I  have few comments:

1) Citation is not placed properly in the text. This makes the manuscript unreadable and immpossible to check the refferences.

2) line 83 - the procedure was the same in each patient? For me it is hard to believe that 3 needle aspirations were performed in each patient.

3) Whether all obtained samples were diagnostic?

4) line 92 - when you mention "tumor specimen" you mean the primary tumor sample? Was it obtained in each patient?

Reviewer 2 Report

Comments
Manuscript ID:
diagnostics-1851896

Tittle:
Prognostic impact of EBUS TBNA for lung adenocarcinoma patients with postoperative recurrences

This is a retrospective study about the prognostic impact of endobronchial ultrasound with transbronchial biopsy (EBUS TBNA) in recurrent lung adenocarcinoma (after radical surgical resection)

Many patients develop recurrence despite curative surgery for non-small cell lung cancer (NSCLC). The identification of the correct candidates for surgery and factors related to recurrence following surgery are very important to guide the administration of adjuvant therapies.

EBUS is the recommended initial procedure for mediastinal nodal staging in patients with NSCL and the precise nodal staging can have an impact in patients’ survival.

Besides the potential originality and interest of this article it has many methodological problems to be considered for publication. The text presented is not completely clear.

-       The methods are not clearly described:

o   It is not mentioned since when EBUS was available in the Thoracic Surgery Division;

o   Did all patients included in this study have the opportunity to perform EBUS?

o   It is not completely clear when was EBUS performed: after radical resection (for diagnosis and staging of recurrent Adenocarcinoma) or before surgery (for diagnosis and staging of primary tumor) – knowing clearly this information is crucial for the interpretation of all the results

o   It is not mentioned how many patients performed PET

o   What were the precise indications to perform EBUS diagnosis and staging (only patients with CT lymph nodes > 10 mm and/or SUV max > 3.5)?; was there an evolution for EBUS indications from 2002 until 2021 in this Thoracic Surgery center (according to the evolution of EBUS guidelines)? It would be expected to have more patients in the EBUS group 

-       Patients were divided in 2 groups (recurrent adenocarcinoma without EBUS / recurrent adenocarcinoma with EBUS); but it is difficult to compare these two groups because

o   Small sample size; very different numbers in the two groups n=218 (non EBUS) vs n=14 (EBUS)

o   Indications for EBUS mediastinal staging changed a lot since 2002 (time of the beginning of this study); considering nowadays recommendations all patients with a central tumor should have been submitted to EBUS (in this study 76 patients with central tumor did not perform EBUS);  

o   We have no information about how many patients in each group performed PET

o   Patients received different treatment regimens

These are not only limitations of the study (as it is said in pag 7) but influenced the results of the study not allowing precise conclusions

-       Results are presented but not discussed or explained:

o   What is the authors explanation for a better survival in patients who performed EBUS-TBNA

o   What is the authors explanation for a higher SUVmax of recurrent tumor in patients who performed EBUS-TBNA

o   What is the authors explanation for more dissected limph nodes in patients who performed EBUS-TBNA

o   What is the authors explanation for: “patients with mutated EGFR got  significantly better survival than patients with wild type EGFR in group without EBUS

o   What is the authors explanation for: “impact of EBUS TBNA to patients with wild type EGFR showed no significant differences with mutated EGFR.

-       A discussion about ROSE with self-citation not relevant for the present paper (pag. 6 lines 182 – 189)

-       Non concordant data presented:

o   number of mediastinal lymph nodes stations biopsied per patient (1.7  pag.  3; 2.3 pag. 7)

o   all EBUS_TBNA procedures were performed by two experienced pulmonologists (pag. 3; line 138); EBUS-TBNA in our study was performed by experienced thoracic surgeons (pag. 8; line 284) 

-       Sentences difficult to interpret such as:

o   In the univariate regression analysis, SUVmax of recurrent tumors, number of dissected lymph nodes and survival were significant predictors for the prognostic impact of EBUS TBNA.

o   But the use of EBUS TBNA could help no significant difference between patients with mutated EGFR and wild type.